# A prospective single-masked, non-inferiority, parallel-group randomized controlled trial of the efficacy of a ChatGPT-based AI chatbot to improve Boston bowel preparation scores for colonoscopy preparation: A trial protocol

**Nabil Mohammad Azmi**[1]*, **Muhammad Irfan Abdul Jalal**[2]*, **Siti Hamizah Mohd Ashar**[1], **Muhammad Irfan Mohd Nazri**[1], **Young Jie**[1], **Nagulan Ganeson**[1], **Joane K. Augustine**[1], **Yew Sheng Qian**[3]

1 Department of Surgery, Faculty of Medicine, Hospital Canselor Tuanku Muhriz, Universiti Kebangsaan Malaysia, Kuala Lumpur, Malaysia, 2 UKM Medical Molecular Biology Institute (UMBI), Hospital Canselor Tuanku Muhriz, Universiti Kebangsaan Malaysia, Kuala Lumpur, Malaysia, 3 Department of Public Health Medicine, Faculty of Medicine, Universiti Kebangsaan Malaysia, Kuala Lumpur, Malaysia

* nabil@ukm.edu.my (NAM); irfan.abduljalal@ukm.edu.my (MIAJ)

## Abstract

Artificial intelligence (AI) is transforming healthcare through tools like large language model chatbots. AI chatbots can simulate human conversation, provide personalized information, and interact with patients in real time. Their ease of use and conversational interface make them attractive for healthcare education, especially in resource-limited settings. We propose a prospective, single-masked, randomized controlled trial to evaluate whether an AI-based chatbot (ChatGPT) is non-inferior to standard counseling in terms of patients' adherence to pre-colonoscopy bowel preparation instructions and thus enhance the Boston Bowel Preparation Score (BBPS). Patients undergoing colonoscopy ($n_{total}$ = 96) will be randomized to ChatGPT 4.0 Large Language Model (LLM)-aided Colonoscopy Counseling ($N_{ChatGPT}$ = 48) or standard counseling ($n_{sc}$ = 48) arms at a 1:1 ratio using a central block randomization scheme of varying block sizes. In the first group, participants will interact with ChatGPT 4.0 for bowel preparation counseling before colonoscopy, whilst the second group will receive standard counseling from trained clinicians. Only the outcome assessors will be masked to the intervention allotment. The primary endpoint is the BBPS, assessed for non-inferiority. Secondary endpoints are patient anxiety (DASS-21) and patient satisfaction assessed using DASS-21 and PSQ-18 questionnaires, respectively and the findings will be reported descriptively with two-sided 95% confidence interval and any p-values will be considered exploratory without multiplicity adjustment. The primary endpoint data will be analyzed using the intention-to-treat (ITT) analysis and non-inferiority framework based on the analysis of covariance (ANCOVA) to control the confounders (age, gender (male as the risk factor), prior

**Data availability statement:** Deidentified research data will be made publicly available when the study is completed and published.

**Funding:** This study is financially sponsored by the Faculty of Medicine Fundamental Grant, Universiti Kebangsaan Malaysia (Project code: FF-2025-112).

**Competing interests:** The authors have declared that no competing interests exist.

**Abbreviations:** AI, Artificial Intelligence; BBPS, Boston Bowel Preparation Score; CRC, Colorectal Cancer; ADR, Adenoma Detection Rate; DASS-21, Depression Anxiety and Stress Scale-21; PSQ-18, Patient Satisfaction Questionnaire-18.

colonoscopy experience, colonoscopy indication, and baseline constipation score). The results will be compared with the findings based on the per-protocol (PP) analysis as part of the sensitivity analysis. The protocol adheres to SPIRIT 2025 and the SPIRIT-AI extension guidelines to ensure comprehensive reporting of this AI-based intervention. This trial has received ethics approval and the trial protocol has been registered with the clinicaltrials.gov registry (NCT06905782).

## Introduction

Colonoscopy is the gold standard for colorectal cancer (CRC) screening and detection of adenomas. Its life-saving potential depends critically on high-quality bowel preparation, as poor preparation can reduce the adenoma detection rate (ADR) and necessitate repeat procedures [1,2]. Inadequate preparation also increases costs and may delay diagnosis of precancerous or malignant lesions, thereby compromising patient outcomes [3]. Previous research demonstrated that up to one-third of colonoscopies have suboptimal preparation, with rates between 6.8% and 33% [3]. Common factors associated with poor bowel preparation include patient non-compliance, misunderstanding of instructions, and procedural anxiety, often influenced by socio-demographic variables such as older age and lower education [4]. Efforts to improve preparation have focused on patient education and support.

Adherence to clear bowel preparation instructions is crucial for colonoscopy success. However, ensuring patient compliance remains challenging due to barriers such as limited health literacy and anxiety about the procedure [4]. Inadequate preparation often leads to failed colonoscopies, delayed diagnoses, and increased healthcare burden. Conventional counseling by healthcare providers has limitations in reaching and engaging all patients effectively.

Artificial intelligence (AI) offers innovative solutions to augment patient education. AI systems have achieved remarkable success in medical image analysis, for example, with deep convolutional neural networks (CNNs) improving bowel cleanliness scoring (e.g., ENDOANGEL) [5]. Yet, most image-based AI tools do not directly enhance patient communication. In contrast, large language models like ChatGPT can engage patients in natural conversation. ChatGPT (based on GPT-4) has demonstrated proficiency in medical text processing and can improve doctor–patient dialogue [6]. Its multimodal abilities also extend to image recognition, suggesting future versatility in patient interaction.

AI chatbots are emerging in healthcare for tasks including triage, education, and counseling [7–16]. By simulating a human conversational partner, chatbots can deliver standardized and personalized guidance on a large scale. They are accessible via widely available digital platforms (smartphones, computers) and can overcome provider shortage by supplementing routine care [7–9,12–16]. Chatbots can also actively engage users who have poor health literacy or limited access by using clear, jargon-free language and continuous availability [12–15]. Two recent systematic reviews found that conversational agents generally improve patient engagement

and outcomes, for example, 75% of studies reported positive effects on usability and adherence [10,17]. In healthcare settings, chatbots have shown promise in increasing adherence to pre-procedure protocols through interactive guidance, and they can help alleviate patient anxiety by providing supportive information.

Recent work has examined ChatGPT-4 in a different part of the bowel-preparation pathway, namely, as a stand-alone image grader of bowel cleanliness on colonoscopy frames using the Boston Bowel Preparation Scale (BBPS). In that concordance analysis, ChatGPT-4 labelled fewer frames as adequately prepared than human endoscopists (62.9% vs 89.3%), with slight agreement on adequacy ($\kappa = 0.099$, $p < 0.001$) and moderate concordance for raw BBPS scores ($W = 0.554$, $p = 0.036$) [18]. However, our trial addresses a distinct question: whether an LLM-based patient-education adjunct can improve adherence to standard preparation. In our study, BBPS is scored exclusively by masked endoscopists during colonoscopy, and the chatbot does not perform any image interpretation or outcome assessment. To date, most evidence has come from psychiatric [19] or medical education studies [20], with limited data on procedural preparation such as colonoscopy. Studies noted gaps in safety, privacy, and bias assessment in many AI applications [17]. To address this concern, this trial incorporates best practices for AI interventions. The protocol explicitly follows the Standard Protocol Items: Recommendations for Interventional Trials (SPIRIT) 2025 guidelines and the SPIRIT-AI extension for AI-based clinical trials, ensuring transparent reporting of the AI system, input/output handling, human–AI interaction, error monitoring, and performance evaluation [21,22].

We will compare standard bowel preparation for colonoscopy counseling with an AI chatbot–augmented approach, together with conventional healthcare worker counseling. We hypothesize that patients who interact with the ChatGPT-based counselor will have better bowel cleanliness (based on BBPS scores) compared to controls. The study will also exploratorily compare anxiety and satisfaction of trial participants receiving these two different interventions and any AI-related safety issues. Here, we present the trial design and methods.

## Methods

### Study design

This is a prospective, single-center, single-masked (outcome assessor), two-parallel arm, non-inferiority randomized controlled trial. The primary goal is to assess whether AI chatbot counseling improves bowel preparation quality based on BBPS score for elective colonoscopy. Eligible patients will be randomized (block randomization, block size = 4) in a 1:1 ratio to either the AI intervention group or the control group receiving standard instructions. Endoscopists, who will score the BBPS during colonoscopy, will be masked to group allocation. The protocol adheres to SPIRIT 2025 guidelines and has received ethical approval from the Universiti Kebangsaan Malaysia (UKM) Research Ethics Committee (Ref: UKM PPI/111/18, Date of approval: 14th Mar 2025). Trial registration with ClinicalTrials.gov has been approved on 01/04/2025 (ID: NCT06905782).

Table 1 shows the SPIRIT 2025 schedule of participant enrollment, administration of interventions and evaluations of trial endpoints. The structured, summarized details of the trial design, based on the World Health Organization (WHO) Trial Registration Dataset, are given in Table 2. The planned participant data collection is summarized in Table 3. Written informed consent will be obtained from each study participant by the trained study recruiters. All participants will be covered by private insurance schemes as a means of compensation in the event of adverse events (including death) occurring related to the procedures. The participants will not be involved in the design and conduct of the trial since the trial will not involve any patient advocacy groups.

### Study setting

The trial will be conducted at Hospital Canselor Tuanku Muhriz (HCTM), the teaching hospital of UKM in Cheras, Kuala Lumpur. Patient recruitment will be carried out between 01/04/2025 and 31/07/2025.

**Table 1. Schedule of participant enrollment, administration of interventions and assessments of study outcomes for this trial (adapted from SPIRIT 2025 guideline).**

| Time points | Trial Period | | | | |
|---|---|---|---|---|---|
| | Enrollment | | Post Randomization | | During colonoscopy |
| | -t$_i$ to 0 | 0 | t$_1$ | t$_2$ (after intervention) | t$_2$ |
| **Enrollment** | | | | | |
| Screening for trial eligibility | X | | | | |
| Informed consent | X | | | | |
| Baseline participant demographic data collection | X | | | | |
| Randomization | | X | | | |
| **Interventions** | | | | | |
| ChatGPT-delivered instructions for colonoscopy preparation (Intervention of interest) | | | X | | |
| Standard instructions for colonoscopy preparation (Comparator) | | | X | | |
| **Assessments** | | | | | |
| Boston Bowel Preparation Scores (Bowel Preparation Quality) | | | | | X |
| DASS-21 score (Anxiety) | | | | X | |
| PSQ-18 score (Patient's satisfaction) | | | | X | |
| Satisfactory bowel preparation status (Bowel Prep Scores ≥2 in all three colon segments) | | | | | X |
| Colonoscopy quality metrics | | | | | X |

## Eligibility criteria

Individuals who wish to take part in this study must meet several inclusion criteria. They need to be at least 18 years old and scheduled for a first-time or repeat elective colonoscopy at HCTM. Besides, participants should be comfortable using a smartphone, tablet, or computer to interact with a web-based system, have basic digital literacy and have personal internet access (provided by HCTM if needed). We require participants to have basic digital literacy in order to interact with the chatbot interface. For participants who are otherwise eligible but lack a personal device, the trial will provide a loaner tablet and a short onboarding session (10 minutes or less) to enable participation. The provision of loaner devices aims to mitigate but not eliminate selection bias; the trial remains a pragmatic first-phase evaluation among individuals who can use a mobile/tablet interface. They also need to be able to understand and communicate in either English or Bahasa Malaysia, as the chatbot used in the study will rely on these languages. Lastly, they must be prescribed a standard bowel preparation regimen by their attending doctor, without any special adjustments.

At the same time, the following individuals will be excluded from the trials: those with significant cognitive problems, such as dementia or severe neurological conditions, that affect their ability to follow instructions, patients who require a customized bowel preparation, for example due to chronic kidney disease, inflammatory bowel disease, or similar conditions, those with severe anxiety disorders or other psychiatric illnesses that could affect how they experience or report anxiety and individuals who are not able or do not consent to study participation.

## Randomization and masking

After eligibility screening and informed consent, participants will be randomly allocated to the AI chatbot arm or control arm. A computer-generated random sequence (blocks of 4) will be created by the trial statistician (MIAJ) for assigning participants to the allotted interventions in a 1:1 ratio and this will be stored in the REDCap system. All other study personnel will not have access to the randomization sequence to maintain allocation concealment. Recruitment will be mainly

**Table 2. Components of the WHO Registration Data Set. Extracted from the ClinicalTrials.gov registry and adapted from the SPIRIT 2025 guideline.**

| Components | Details |
|---|---|
| **Primary registry and trial identifying number** | Clinicaltrials.gov registry<br>NCT06905782<br>Weblink: https://clinicaltrials.gov/study/NCT06905782?id=NCT06905782 |
| **Date of registration in the ClinicalTrials.gov registry** | 01-04-2025 |
| **Secondary Identifying Number** | JEP-2025–035 |
| **Universal Trial Number (UTN)** | N/A |
| **Source of monetary and material support** | Faculty of Medicine, Universiti Kebangsaan Malaysia (UKM) |
| **Primary Sponsor** | Faculty of Medicine,<br>Hospital Canselor Tuanku Muhriz (HCTM),<br>Universiti Kebangsaan Malaysia (UKM),<br>Jalan Yaacob Latif, Bandar Tun Razak,<br>56000, Cheras,<br>Kuala Lumpur, Malaysia.<br>Email: sepukm@ukm.edu.my |
| **Secondary Sponsor** | N/A |
| **Contact for Public Queries** | Dr Nabil Mohammad Azmi<br>Email: nabil@ukm.edu.my |
| **Contact for Scientific Queries** | Dr Nabil Mohammad Azmi<br>Email: nabil@ukm.edu.my |
| **Public Title** | ChatGPT for Bowel Preparation Counseling Before Colonoscopy |
| **Scientific Title** | Prospective Single-Blinded Randomized Control Trial on the Effectiveness of Using Large Language Model Artificial Intelligence Chatbot to Improve Boston Bowel Preparation Score (BBPS) for Colonoscopy Preparation |
| **Countries of Recruitment** | Malaysia |
| **Health Condition** | Suspected colorectal adenoma and colorectal cancer |
| **Interventions** | **Experimental Arm:** ChatGPT 4.0 Large Language Model (LLM)-aided Colonoscopy Counseling Randomized participants will be interacting with ChatGPT 4.0 for bowel preparation counseling prior to colonoscopy.<br>**Control arm:** Standard Colonoscopy Counseling Randomized participants will receive standard counseling delivered by trained medical personnel for bowel preparation prior to colonoscopy |
| **Key Eligibility Criteria** | <u>Inclusion Criteria:</u><br>1) All scheduled colonoscopy with indication<br>2) Adequate digital literacy<br>3) Adequate language literacy with Malay and English language<br><u>Exclusion Criteria:</u><br>1) Patients with memory impairment due to previous stroke, dementia or Alzheimer's disease 2) Diagnosed with clinical anxiety |

*(Continued)*

**Table 2.** (Continued)

| Components | Details |
|---|---|
| **Trial Type** | Purpose: Diagnostic<br>Allocation: Randomized Controlled Trial<br>Framework: Non-Inferiority<br>Masking: Single Masked [Outcome assessors (Endoscopists)]<br>Assignment: Parallel<br>Type of endpoint: Efficacy<br>Phase: NA (Diagnostic Trial) |
| **Date of First Enrolment** | 01/04/2025 |
| **Sample Size** | 43 participants per arm ($n_{total}$ = 86 participants)<br>Including 10% drop-out: 48 participants per arm ($n_{total}$ = 96 participants) |
| **Recruitment Status** | Recruiting |
| **Primary Outcome** | **Bowel Preparation Quality:** Measured by the Boston Bowel Preparation Score (BBPS) [range for total scores: 0 (poor) to 9 (excellent)) taken from three colonic segments (right, transverse and left colon; each segment will be scored using BBPS scores of between 0 (poor) to 3 excellent] during colonoscopy. |
| **Secondary Outcomes** | 1) **Patient Anxiety Score:** Measured by the English and Malay-translated and validated Depression Anxiety and Stress Scale-21 (DASS-21). Anxiety level will be assessed on the day of the procedure (pre-colonoscopy), reflecting state anxiety after the intervention versus baseline expectations.<br>2) **Patient Satisfaction:** Measured by the English and Malay-translated and validated Patient Satisfaction Questionnaire-18 (PSQ-18) immediately after the intervention (on the day of colonoscopy). This captures satisfaction with the preparation process and information received.<br>3) **Adequate Bowel Preparation Rate:** Proportion of patients achieving adequate BBPS (score ≥2 in all three colonic segments)<br>4) **Other Colonoscopy Quality Metrics:** Including polyp detection rate, adenoma detection rate and cecal intubation rate, to contextualize preparation quality. |
| **Ethics Review** | **Status:** Approved<br>**Approval Date:** 14-03-2025<br>**Approval ID:** UKM PPI/111/18,<br>**Contact:** UKM Research Ethics Committee |
| **Data Sharing Statement** | Anonymized and de-identified participants' clinical and trial outcome data will be shared through the Harvard Dataverse Repository (https://dataverse.harvard.edu/). The full statistical codes used for trial data cleaning, transformation and analysis will be made available on the repository. |

N/A: Not Available; WHO: World Health Organization

**Table 3. Relevant participant demographics will be collected during the trial.**

| Items | Descriptions |
|---|---|
| **Section 1: Patient Demographics** | |
| Name | |
| Age (in years) | |
| Gender | |
| Body Mass Index (BMI, in kg/m$^2$) | |
| Profession/ Occupation | |
| Digital Literacy Level | Basic/Intermediate/Advanced |
| Comorbidities | Diabetes Mellitus/Hypertension/Chronic Kidney Disease/Ischemic Heart Disease<br>Others: |
| Relevant medical history | Previous history of constipation (Rome's Criteria)/ Previous history of diarrhea<br>Others: |
| Prior Colonoscopy History (if yes, please provide the year of most recent colonoscopy) | Yes/No<br>Year of most recent colonoscopy: |
| Smoker | Yes/No |
| Alcohol drinker | Yes/No |
| **Section 2: Bowel Preparation** | |
| Dietary restriction before colonoscopy | Yes/No |
| Completion of laxative (Fortrans™) as instructed | Yes/No<br>If No state the reasons: |
| **Section 3: Colonoscopy Details** | |
| Indication for colonoscopy | |
| Dosage of sedation (IV Midazolam in mg & IV Pethidine in mg) | |
| Completion of colonoscopy | Complete/Incomplete<br>If Incomplete, state the reason: |
| Time taken (in minutes) | |
| Withdrawal time (in minutes) | |
| Diagnosis | |
| Bowel preparation method | |
| Boston Bowel Preparation Scores (BBPS) | |
| Finding(s) | |
| Intervention during colonoscopy | (Yes/No) If yes, state the indication:<br>• Biopsy<br>• Polypectomy<br>• Others: _________________ |
| Histopathology examination (if available). | Benign/Malignant. |
| Adverse event | Yes/No<br>If yes, state the event: |
| Baseline Anxiety Level (DASS-21 score) | |

performed by the principal investigator (NMA). The REDCap system will only be used by the study personnel to allocate participants to trial interventions after participants have consented to trial participation. The trial statistician (MIAJ) will neither be involved in participant recruitment nor in the administration of the allotted intervention.

Participants and other study personnel will not be masked to the assigned interventions, but the primary outcome will be objectively assessed by endoscopists who will be masked to the interventions received by the participants. Complete masking of participants and staff is not feasible since participants are inherently aware of whether they interact with an AI chatbot or a clinician, and staff facilitating the counseling or questionnaires necessarily know the intervention delivered. However, staff influence will be minimized through strict adherence to standardized hospital counseling protocols (control arm) and scripted questionnaire delivery (ChatGPT arms). Importantly, the BBPS is objectively evaluated by endoscopists who are masked to allocation, preserving the integrity of the primary endpoint evaluation. The endoscopists and participants will be instructed to maintain minimal conversation to prevent unmasking. No emergency unmasking is anticipated since we do not expect any significant harm to be associated with the interventions received by the participants.

## Interventions

**Control Arm (Standard Counseling).** Participants in the control group will receive routine pre-procedure counseling as per hospital protocol. This includes written instructions (as a brochure) and/or face-to-face explanation by trained healthcare staff about dietary restrictions and bowel preparation steps. The counseling covers key points on diet modification and laxative use (Polyethylene glycol 4000 (Fortrans™); DCH-Auriga, Malaysia), where one sachet of Fortrans™ will be mixed with one liter of water and taken at 6:00 PM and 8:00 PM the night before, and another 1 liter at 6:00 AM on the colonoscopy procedural day. The content and dosing regimen were developed with reference to the European Society of Gastrointestinal Endoscopy (ESGE) Guideline Update 2019 on bowel preparation for colonoscopy [23] and the Malaysian Ministry of Health (MOH) Clinical Practice Guidelines on colorectal cancer screening [24]. Our regimen specifies that the final sachet should be consumed 4–5 hours before the procedure, completed at least 2 hours before colonoscopy. This adaptation reflects local institutional policy and MOH recommendations, while remaining aligned with ESGE's general principles of split dosing and minimum fasting intervals. This is further corroborated by evidence that modest variations within this range do not compromise bowel cleanliness or adenoma detection [25,26]. Standard practice may include answering patient questions in person but does not involve any AI tools.

**Intervention arm (AI-ChatGPT Chatbot Counseling).** Participants in the intervention group will receive a supplementary counseling session via an AI chatbot powered by ChatGPT. The chatbot content is customized for colonoscopy preparation and is based on reputable guidelines from the ESGE and MOH recommendations. Each participant will first be given a personalized script (electronic or printed) that contains their name, age, relevant comorbidities, current medications, and scheduled procedure date. This script also summarizes the standard preparation protocol for clarity. Participants choose their preferred language (English or Malay), and the script is provided in that language.

After reviewing the script, participants will access the ChatGPT-based chatbot on a hospital-provided laptop (or personal device) in a private area. The chatbot is implemented via the OpenAI web interface (using ChatGPT version 4.0). The chatbot's parameters are fixed (temperature = 1.0 for creative and empathetic responses, max tokens = 200,000) and each user's query will be submitted as a new session to prevent context carryover. A temperature parameter of 1.0 was chosen since our pilot testing showed that lower values (0.0–0.7) produced rigid, sometimes confusing phrasing that reduced patient comprehension, while 1.0 maintained natural dialogue without compromising accuracy of instruction for bowel preparation. Besides, all dosing and safety-critical instructions (dietary restrictions, purgative timing, fasting intervals, sedation precautions) are delivered using templated, pre-specified messages to ensure consistency across participants. The temperature parameter affects only conversational phrasing and clarifications, selected to improve patient comprehension and engagement.

To ensure the participants received consistent guidance throughout the trial, the ChatGPT used in this trial runs on a locked local deployment finalized before the trial starts to eliminate variability due to external model updates. Core educational content, including dietary restrictions, purgative timing, and fasting intervals, is implemented as templated outputs to

ensure that all participants receive identical evidence-based instructions. The language model delivers this content conversationally and provides clarifications, but does not alter the dosing regimen. All interactions are stored as transcripts with date and time stamps for monitoring.

The chatbot interaction proceeds as follows: the participant inputs their personalized script, and then the chatbot provides step-by-step guidance on dietary restrictions (e.g., clear fluid allowance, solid food cutoff) and bowel preparation (laxative timing and use). The chatbot also offers motivational encouragement and reinforcement messages. Participants may ask the chatbot additional questions about the procedure and preparation; the chatbot will answer within predefined safety limits (e.g., it can clarify instructions but will not give medical diagnoses). No personal identifiable information (PII) is shared with ChatGPT since all interactions are captured via anonymized user inputs. If any user input is incomplete or unclear, the system's input validation will prompt for clarification. Clinic staff are available to assist with technology, but will not interject into the chatbot responses. Each chatbot session is expected to last 15–30 minutes.

The instructions for bowel preparation delivered by the ChatGPT chatbot will be developed through an iterative process: Clinical guidelines (ESGE, MOH) will be used to construct initial patient questions and corresponding answers. We will subsequently employ ChatGPT to draft dialogue scripts in both English and Malay. Multiple refinements ensured localization (e.g., replacing "pencahar" with "pelawas" in the Malay language, specifying local timing for laxative doses at 6 PM/8 PM and 6 AM/ 9 AM) and alignment with institutional practice. The final script will be structured into conversational sections (Introduction, Dietary Preparation, Bowel Preparation, Mental Preparation, Risks & Safety, Day of Procedure) to mirror typical patient concerns. Clinicians will review the content for accuracy, cultural appropriateness, and readability before deployment. Thus, ChatGPT's knowledge base consists of standardized, evidence-based bowel preparation protocols, with patient-friendly language. The readability of the responses generated by ChatGPT will be assessed using the Flesch-Kincaid Grade Level score and the Gunning-Fog index [27,28].

Those who cannot complete the structured input forms or who submit incomplete, non-English/non-Malay, or ambiguous responses will be identified during enrollment and will be referred to standard counseling instead (data from such cases will not enter AI analysis). The chatbot will not collect PII, and responses longer than 500 words (approximately between 800 and 1000 tokens) will be truncated to fit processing constraints.

In this trial, ChatGPT is used only to support standard care by providing text-based bowel-preparation advice. Clinicians retain full responsibility for all decisions and will decide when in-person counseling is needed. If a patient raises questions that ChatGPT cannot answer, or if the information provided is not adequate for safe decision-making, the clinician in charge will step in and deliver conventional, in-person counseling.

**AI System Infrastructure and Privacy.** The chatbot is hosted on a secure cloud platform with strong data protection. Access is restricted to trial participants and authorized study personnel through authentication controls. All data transmissions are encrypted end-to-end. The ChatGPT system logs interaction data (timestamps, user inputs, and chatbot responses) for analysis, but no identifiable patient information is ever stored. Input restrictions (English/Malay text only, structured formats) ensure compatibility with the model. Responses exceeding the token limit are automatically truncated or rephrased by the system.

Clinical sites will have internet-enabled devices (computers, tablets, or smartphones) available in private rooms for supervised interactions if needed. Printed or digital instructions will orient participants on using the chatbot platform. Patients may also access the chatbot remotely via their own devices to review the information later. Technical support will be available to troubleshoot any access issues.

**AI error taxonomy, thresholds and escalation.** All chatbot interactions will be logged and reviewed according to a pre-specified error taxonomy. Errors are classified as Critical, Major or Minor (definitions below). A weekly random sample of 10% of sessions will be reviewed by trained clinicians and review sampling will expand to 25% if thresholds are exceeded. Escalation rules are:

1. Critical error: any instance of advice that could cause direct physical harm (e.g., wrong dosing interval for the laxatives leading to electrolyte imbalance; instruction to discontinue essential medication without context) will immediately suspend the affected module and trigger urgent review by the Safety Officer and Data Monitoring Committee (DMC).

2. Major errors: defined as incorrect or misleading clinical advice that could materially reduce bowel preparation adequacy (e.g., wrong laxative timing exceeding 6 hours). If 1% or more of audited sessions in a rolling 2-week window contain Major errors, root cause analysis will be performed, chatbot prompts/content will be corrected, and audit sampling will increase to 25% until the error rate falls below the threshold.

3. Minor errors: non-material issues such as ambiguous wording, style, or formatting. If more than 5% of audited sessions contain minor errors in two consecutive reviews, content will be refined and monitored more closely.

This structured governance approach aligns with international AI safety recommendations, including WHO and SPIRIT-AI/CONSORT-AI guidance [29,30], healthcare AI safety principles [31–33], the National Institute of Standards and Technology (NIST) AI Risk Management Framework [34], and explainability principles for multidisciplinary evaluation of AI in health [35].

**Algorithm Details and Updates.**  The AI chatbot uses OpenAI's ChatGPT-4.0 model accessed using the Application Programming Interface (API). The deployed model version and system parameters (temperature, token limit) will be documented. If OpenAI releases updates during the trial period, we will record any change in model version and assess for consistency. Since ChatGPT is **a proprietary** model, its internal weights are not modifiable; we rely on configuring prompts and contexts. Our focus is on standardizing inputs and validating outputs to achieve reproducibility.

**Human–ChatGPT Interaction and Expertise.**  The intervention is designed for direct patient use without real-time expert oversight. Participants need only basic digital literacy. Healthcare staff will introduce the chatbot and assist with any technical difficulties or initial login. They will not censor or alter the chatbot's responses, but will be available to clarify instructions if the participant requests additional explanation. No specialized training in AI is required for staff or participants.

**Error Monitoring and Performance Evaluation.**  Given the novel AI-ChatGPT intervention, we will systematically monitor chatbot performance. The study will classify any AI-generated errors (e.g., incorrect guidance, omissions, ambiguous responses) by frequency and type. Automated system logs will detect anomalies or repeated errors in output patterns. Additionally, at least 10% of chatbot sessions (randomly selected) will be reviewed by clinicians to assess the accuracy and appropriateness of the guidance. Participants can also flag any confusing or incorrect responses, which will be recorded. We will calculate error rates (e.g., percentage of responses flagged) and perform qualitative case reviews to identify common issues.

To mitigate safety risks, the chatbot's knowledge base is restricted to evidence-based bowel prep content. Potential hallucinations (fabricated or unsafe advice) are minimized by configuring the prompt context to clinical guidelines and by implementing real-time checks: if the chatbot's output falls outside established instructions or is contradictory, it will be logged and corrected offline. In practice, if a response appears clinically inappropriate (e.g., suggesting contraindicated actions), the participant will be directed to human counseling immediately. An escalation protocol ensures that any ambiguous or safety-related queries are referred to staff. These measures are intended to catch and correct AI errors before they impact patient care.

**Bias, Explainability, and Reproducibility.**  We will actively address common AI concerns. The chatbot's training and prompts are based on standardized clinical protocols to reduce hallucinations or omissions. Apart from those, during the prompt engineering phase, we will also use the shortest possible prompts based on formal languages with maximum use of concrete words representing tangible objects to reduce ChatGPT's hallucination rate [36]. Besides, the ChatGPT system is tested to operate within defined knowledge boundaries (preventing speculative answers). We will conduct periodic audits of AI advice across demographic subgroups to check for any bias (e.g., differences in style or content by

patient age or language). If any bias or disparity is found (for example, if the AI inadvertently gives different instructions for Malay versus English outputs), we will adjust the model prompts accordingly.

The interaction framework is standardized: inputs are structured (same script template) and outputs follow fixed templates (sections on diet, instructions, etc.). This consistency enhances reliability and reproducibility. We will document the exact prompt templates, session settings, and output formats to enable replication. In addition, patient-facing language is kept clear and conversational to aid explainability. Participants can request simpler explanations from the chatbot if needed.

**Data Protection and Privacy for ChatGPT-based intervention development.** All AI interactions are designed with privacy safeguards. No PII is entered into the chatbot system. Data transmissions are encrypted, and the platform enforces access controls. Participant responses are de-identified before storage; we will use de-identification techniques to ensure that logged transcripts cannot be traced back to individuals. All data analysis will use anonymized IDs.

**Reproducibility and Access to Intervention Details.** ChatGPT 4.0 is a proprietary model not available for modification or distribution. Therefore, to support reproducibility, we will fully describe the intervention's configuration: model version, system parameters, prompt designs, and supervision procedures. Study findings, including the scripts used and performance results, will be published. While the chatbot code cannot be shared, researchers with appropriate licenses could replicate the approach by following our documented methods. Anonymized interaction data (without PII) may be made available to other researchers upon reasonable request and institutional approval.

**Assessments of Intervention Adherence.** Adherence will be assessed using a triangulated approach: (a) participant self-report with timestamps (intervention: chatbot confirmation logs; control: electronic diary/phone confirmation); (b) nurse-verified time of last oral intake on admission; and (c) cross-checking where discrepancies exist.

Adherence categories are pre-specified, consistent with ESGE guidelines and published adequacy thresholds [23,37]:

- Full adherence: all doses consumed and final dose initiated between 4 and 6 hours before procedure start and completed 2 hours or more before procedure.

- Partial adherence: Single missed dose or final dose outside the 2-to-6-hour window but within 12 hours.

- Non-adherent: More than one missed dose or final dose initiated more than 12 hours before procedure.

The per-protocol population comprises participants meeting full adherence. Adherence rates will be reported by arm and incorporated in sensitivity analyses, following recommendations for transparent adherence reporting [38].

**Trial endpoint assessments.** The primary endpoint of this study is the quality of bowel preparation, assessed using the BBPS during colonoscopy [39,40]. This scoring system evaluates three segments of the colon, right, transverse, and left, with each segment scored from 0 to 3, resulting in a total score ranging from 0 (indicating poor preparation) to 9 (indicating excellent preparation). Colonoscopy and BBPS scoring will be performed by two consultant-level endoscopists, each with at least five years of independent colonoscopy practice and have carried out at least 200 colonoscopies per year. Prior to trial commencement, all participating endoscopists undertook a structured calibration exercise involving independent scoring of recorded colonoscopy videos using the BBPS, followed by consensus discussion until concordance was reached. During the trial, inter-observer reliability will be evaluated by secondary review of 10% of procedures using recorded images, with agreement quantified using intra-class correlation coefficients (ICC). Periodic calibration meetings will be held to maintain consistency [39,40]. Each patient's total BBPS score will serve as the primary outcome measure for comparing bowel preparation quality across study arms.

Several secondary endpoints will also be assessed. Patient anxiety will be measured using the anxiety subscale of the Depression Anxiety and Stress Scale-21 (DASS-21), administered on the day of the colonoscopy to evaluate state anxiety following the intervention. Patient satisfaction will be assessed using the Patient Satisfaction Questionnaire-18 (PSQ-18), completed immediately after the procedure to capture perceptions of the preparation process and the clarity of information provided. The DASS-21 and PSQ-18 were selected since both have been validated and widely used instruments with

strong psychometric properties in our setting. We acknowledge, however, that DASS-21 uses a one-week recall frame and may not fully capture very acute anxiety fluctuations on the day of colonoscopy and that PSQ-18 scores could be influenced by residual sedation. We will acknowledge this limitation when interpreting the trial results, though the application of identical instruments across both groups may reduce the risk of bias.

All questionnaires have been translated into Malay and have good properties of inter-rater reliability and validity: DASS-21 for anxiety (Cronbach's α: 0.81 (English) – 0.84 (Malay)) [41,42], and PSQ-18 for satisfaction (Cronbach's α: 0.745 (Malay) – 0.750 (English)) [43,44]. All secondary outcomes are scored on Likert scales for standardization. Anxiety and satisfaction are assessed at the time of the procedure when patient concerns are most salient, providing a realistic gauge of the intervention's impact.

To minimize performance and detection bias, patient-reported outcomes (DASS-21, PSQ-18) will be administered by trained research assistants using a verbatim scripted protocol, without disclosing intervention allocation. Patients will complete questionnaires electronically, ensuring self-entry and eliminating interviewer influence. Research assistants will receive structured training and ongoing monitoring to ensure protocol fidelity. All questionnaires will be administered at prespecified time points in a quiet, standardized environment.

Finally, the proportion of patients achieving adequate bowel preparation, defined as a BBPS score of at least 2 in all three segments, will be calculated to determine overall effectiveness. Additional exploratory outcomes will include standard colonoscopy quality metrics such as the adenoma detection rate (ADR) and cecal intubation rate, providing context for interpreting the impact of preparation quality on procedural performance.

## Adverse event assessments and monitoring

Adverse events (AE) are defined as "an abnormal sign, symptom, laboratory test, syndromic combination of such abnormalities, untoward or unplanned occurrence (e.g., accident), or any unexpected deterioration of concurrent illness" [45]. For serious AE (SAE), this is defined as "adverse events that result in the following outcomes: 1) death; 2) life-threatening AEs; 3) inpatient hospitalization or prolongation of existing hospitalization; 4) a persistence of significant incapacity or substantial disruption [46].

Earlier research has demonstrated that the anticipated adverse events are essentially consistent with the complications typically associated with standard diagnostic and therapeutic colonoscopy procedures, such as abdominal bloating, flatulence, and the rarer but more serious adverse events, such as colonic perforation, infection, and post-polypectomy syndrome. Besides, the ChatGPT utilized in this study is a proprietary large language model that does not involve direct contact with the human body. Therefore, ChatGPT is assumed to have minimal harm to study participants.

The participants will be monitored for 24 hours following colonoscopy for AE and SAE through clinical observations in the daycare ward and through telephone calls once the participants are discharged. All AEs and SAEs will be recorded in the case report form (CRF) and the details include the adverse event's characteristics, the date and time of onset and disappearance and severity of the AEs. AEs related to colonoscopies will be classified using the American Society for Gastrointestinal Endoscopy (ASGE) endoscopic adverse-event lexicon [47] and reported according to ASGE guidance for colonoscopy complications [48]. Severity of AEs will be classified based on the Common Terminology Criteria for Adverse Events (CTCAE) Version 5 [49], restricted to items relevant to colonoscopy and the peri-procedural period, including gastrointestinal bleeding, perforation, abdominal pain, nausea or vomiting, hypotension and hypoxia. This combined framework combines field-standard endoscopy terminology with a widely used graded severity scale to improve consistency, transparency and comparability across studies. Participants experiencing SAEs will be treated via standard clinical management practice. All AEs grade 3 and above will be reported to the UKM Ethics Committee within five business days. On the other hand, SAEs (including Sudden Unexpected Serious Adverse Reactions (SUSAR)) will be reported to the UKM Ethics Committee within 24 hours (expedited reporting). All study participants will be provided sufficient ancillary care according to the standard clinical management protocol for the treatment of each adverse event.

## Sample size calculation

The trial is only powered for the primary endpoint, the BBPS. The sample size is based on detecting a difference in BBPS scores, which was calculated using PS Software version 3.1.2 (Dupont and Plummer, 2014, available from: https://github.com/vubiostat/ps/raw/refs/heads/master/bin/pssetup3.exe). We utilized the information from a similar trial by Zhu and colleagues [33]. The standard deviation of the BBPS in the standard instruction group was chosen to be 1.81, based on Zhu and co-workers [50]. We also assumed a minimum clinically important difference of BBPS between the ChatGPT-guided and standard instruction groups to be at least 1.1 BBPS points for the non-inferiority margin. This was determined based on validation studies of the BBPS demonstrating that differences one point or less do not materially impact adenoma detection rate (ADR) or adequacy classification (total BBPS score of 6 or more with all segments of 2 or more) [39,40]. This threshold also aligns with ESGE recommendations that bowel preparation adequacy should be maintained in at least 85% of cases [23]. The selected margin, therefore, represents a conservative yet clinically meaningful limit.

Using a two-sample t-test (type I error rate (two-sided) = 0.05, power (1-β) = 80%) and allocation ratio (m) of 1:1, 43 patients per group ($n_{total}$ = 86) are required. Considering a 10% attrition rate due to failure to obtain consents in eligible participants, participant withdrawal prior to randomization and others, the sample size is inflated to 48 participants per group ($n_{total}$ = 96). Hence, we plan to screen 96 potential participants for eligibility before randomization.

No formal sample size calculation was performed for anxiety or satisfaction scores due to limited prior data. The planned sample size is thus not powered to detect differences in secondary endpoints, specifically anxiety (DASS-21) and satisfaction (PSQ-18) scores. This approach is in line with the methodological standards for non-inferiority trials, in which the sample size is driven by a single clinically decisive primary endpoint, while secondary outcomes are used to provide supportive and contextual evidence [51–53].

## Participant recruitment strategy and recruitment monitoring

We will recruit eligible participants who are patients scheduled for elective colonoscopy at the Endoscopic unit of HCTM, who will be identified from the weekly endoscopic list. The patient's eligibility will be verified by medically qualified personnel, who must document this in the patient's medical records upon taking consent for colonoscopy. As an additional strategy to ensure adequate patient recruitment, the medical officers at the HCTM surgical outpatient clinics will be notified to inform the research team members if potentially eligible participants are identified during the outpatient appointments.

Based on our institutional audit data (around 250 colonoscopies per month, with approximately 40 percent meeting eligibility and a 50 percent consent rate), we expect to recruit 96 participants within 3 months (maximum 5 months if needed) at a single center. Recruitment progress will be monitored weekly and contingency strategies are in place should accrual rates fall below expectations based on this traffic light criteria:

Green: 24 or more participants recruited and randomized per month, with less than 10% attrition rate.

Yellow: 15–23 participants recruited and randomized per month, with less than 10% attrition rate. Strategies are found to increase the recruitment rate to 24 or more participants per month (green level).

Red: Less than 15 participants recruited and randomized per month, with less than 10% attrition rate. No strategies are found that can result in a sustainable increase in the recruitment rate to the green level.

The trial will be stopped if the recruitment rate falls under the red category for the trial progress criteria. The trial will be cautiously monitored for two months if the recruitment rate is in the yellow category and will be stopped prematurely if the trial recruitment is within the yellow category for two consecutive months. The progress of participant recruitment will be monitored by the Trial Monitoring Group (TMG) consists of the principal investigator and trial statistician. The TMG's recommendation on the trial continuation status will be relayed and discussed with the funder monthly. The recruitment flow of the study participants is summarized in Fig 1.

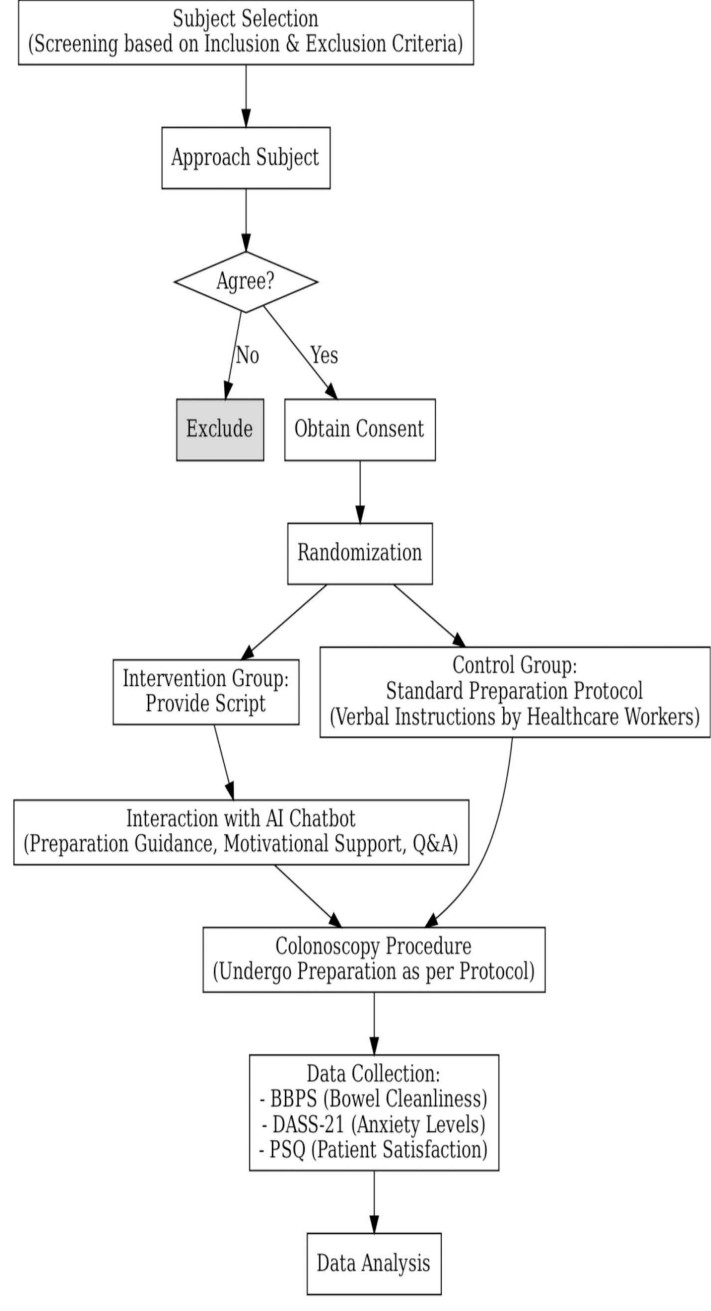

**Fig 1. The planned schematic flow of the clinical trial.**

## Data collection and management

Participants' demographic data (age, gender, BMI, occupation, digital literacy), medical history (comorbidities, medication use, prior GI history, smoking/alcohol), and preparation adherence will be collected by the study investigators. For intervention patients, chatbot interaction logs (questions asked and responses given) will be collected anonymously. Individual participant data (including outcome data) will be entered into a paper-based clinical record form (CRF; available upon

request to the principal trial investigator (NMA)), which will be subsequently entered into a secure SPSS spreadsheet with coded personal identifiers. Only authorized study staff will have access to the data. All data manipulation will be performed using SPSS.

BBPS will be recorded by the endoscopist immediately after colonoscopy using the standard BBPS form. Anxiety and satisfaction questionnaires will be administered by the trained study investigators on the procedure day before sedation. The study investigators will be trained by an independent psychometrician via supervised interviews of potential respondents.

To preserve data quality control, data quality checks (range, consistency) will be performed regularly by the principal investigator and trial statistician. The research personnel collecting the data will also be instructed to evaluate their own performance by checking their individual rate of missing data and implausible responses. Any rectification of errors in data recording is only allowed before the information in the CRFs is transferred into the SPSS spreadsheet. The double data entry procedure will be carried out by two research personnel to prevent errors in data entry.

Besides, missing data for primary and secondary outcomes will be minimized by protocol diligence; all endoscopists will be consistently reminded to record the BBPS score and all colonoscopy quality metrics before performing colonoscopy. Besides, the participants will also be reminded by the trained questionnaire administrators to complete the whole questionnaire and all completed questionnaire forms will be checked for response completion before colonoscopy is carried out.

## Patient confidentiality and data security

The personal identifying information for each participant will be removed from the SPSS spreadsheet. All CRFs will be stored in a designated locked cabinet accessible to the principal investigator and trial statistician only. All electronic datasets will be password protected and encrypted, and the encryption key will be kept by the trial statistician. All CRFs and SPSS datasets will be stored for at least 10 years after the end of the trial and routine data inspections will be made to ensure data readability.

## Statistical analysis plan

Analysis will primarily follow the intention-to-treat (ITT) principle, where all randomized trial participants will be included in the analysis according to their original allocated intervention group. All anonymized and deidentified individual participant data and statistical codes used for the data cleaning, transformation and analysis will be shared via the Harvard Dataverse Repository (https://dataverse.harvard.edu/). Statistical analysis will be conducted using SPSS Version 29 (IBM Corp, Armonk, New York, USA; 2023) or R Version 4.5.1 (R Core Team, Vienna, Austria; 2025) software.

Baseline characteristics will be summarized by group means (standard deviation) or medians (interquartile range) for continuous variables and counts and percentages for categorical variables (e.g., adequate bowel preparation rate). The normality of the continuous variables will be assessed using objective (Shapiro-Wilks test, Fisher's coefficient of skewness [54]) and graphical measures (QQ Plot, Box and Whisker Plot).

Multiple imputations will be carried out if the percentage of missing data is more than 5% per variable under the missing at random (MAR) assumption using multiple imputation by chained equations method implemented on R "mice" package [55]. Following recommended practice, five imputed datasets will be generated, which is considered appropriate for this trial size and avoids spurious variance inflation [56–58]. Results will then be pooled using Rubin's rules, which account for both within-imputation and between-imputation variability to obtain valid statistical inference and reduce Monte Carlo error [57,58].

The primary analysis will then be performed using Analysis of Covariance (ANCOVA), adjusting for age, gender (male as the risk factor), prior colonoscopy experience, colonoscopy indication, and baseline constipation score for the primary trial outcome, BBPS [4,59]. This limited set of covariates was chosen to improve precision without overfitting, in

line with evidence that excessive adjustment in small samples undermines model stability [60,61]. The adjusted mean difference and its 95% confidence interval will be compared with the non-inferiority margin. The non-inferiority margin (Δ) for BBPS score difference will be set at −1.1 points (i.e., the AI-ChatGPT arm is considered non-inferior if the lower bound of the two-sided 95% confidence interval (CI) for the mean BBPS difference AI-ChatGPT – control is greater than −1.1 BBPS points) based on the minimum clinically important difference used for sample size calculation. The primary non-inferiority analysis will use a one-sided t-test at an alpha level of 0.025, testing the null hypothesis that the mean difference in BBPS between the intervention and control groups is less than or equal to −1.1 against the alternative that it is greater than −1.1. The non-inferiority margin of −1.1 BBPS points was selected based on validation studies showing that differences of this magnitude do not compromise adenoma detection rate or adequacy classification (BBPS ≥6 with all segments ≥2) [39,40]. This threshold is also consistent with ESGE guideline quality benchmarks that require at least 85% adequacy in bowel preparation [23]. We will then evaluate the statistical assumptions of ANCOVA (residuals' normality and homoscedasticity, linearity of covariates with the outcome variables, homogeneity of regression slopes) and apply transformations if necessary. Similarly, rates of adequate preparation (BBPS ≥6 overall or ≥2 per segment) will be compared with a chi-square test or Fisher's exact test if the number of cells in the contingency tables with expected counts of less than 5 is more than 20%.

For secondary outcomes, change in anxiety (DASS-21) and satisfaction (PSQ-18) scores, ANCOVA will be employed with statistical adjustment made with baseline anxiety (DASS-21) and satisfaction (PSQ-18) scores. However, for all secondary trial endpoints, the findings will be considered supportive and exploratory only. The results will be summarized with two-sided 95% confidence intervals and any p-values will be treated as exploratory with no multiplicity adjustment, in line with guidance on multiplicity and endpoint hierarchy in clinical trials [52,53,62]. For colonoscopy quality metrics (ADR and caecal intubation rate), comparisons will only be made at the univariable level using the chi-squared or Fisher's exact test. Besides, as a measure of quality control, inter-observer reliability for BBPS scoring will also be reported using intra-class correlation coefficients (ICC) derived from the 10% of procedures undergoing dual review [39]. These results will be presented descriptively as secondary trial outcomes.

For sensitivity analysis, the results obtained based on the ITT analyses will be compared with the findings from the per-protocol (PP) analyses. Besides, if there are trial variables with a more than 5% rate of missingness, the results based on the imputed datasets will be compared with the findings from complete-case analyses. Apart from that, worst-case imputation for missing primary endpoint (control group had the best BBPS scores, AI-ChatGPT group had the worst) will also be employed to assess the robustness of the results. Furthermore, we will also perform an exploratory subgroup analysis that will also compare outcomes between first-time and repeat patients, with a treatment-by-prior-colonoscopy interaction tested descriptively to assess consistency of intervention effects across these groups [63,64]. The conclusion will be based on concordant results (i.e., non-inferiority is observed in all sensitivity analyses) as recommended by the International Council for Harmonization (ICH) E9(R1) statistical principles [65].

For non-inferiority results, they will be reported as point estimates with corresponding 95% CIs and visually presented using the forest plots, with the non-inferiority margins (e.g., −1.1 BBPS points and −10% risk difference) will be clearly indicated as vertical reference lines. One-sided p-value of 0.025 (for primary non-inferiority test) will be employed as the statistical significance thresholds.

## Ethical considerations and safety

The trial will be conducted in accordance with the Declaration of Helsinki and International Council for Harmonization (ICH)'s Principles of Good Clinical Practice (GCP). Participation is voluntary and all patients can withdraw at any time without affecting their care. The chatbot provides non-judgmental guidance, and no medications or invasive procedures are involved in the intervention itself. Should a participant exhibit significant distress or report an urgent medical concern during the chatbot session, staff will intervene according to clinical judgment. Any adverse events

(e.g., severe anxiety spikes) related to the study procedures will be recorded and reported to the ethics committee. Standard clinical practice will continue for all patients (for example, rescheduling a colonoscopy if bowel preparation is insufficient).

### Protocol amendments

Any modifications to this protocol (e.g., changes in chatbot implementation) will be submitted to the ethics committee for approval. The trial registry will be updated with protocol versions. Study progress (enrollment dates, recruitment numbers, withdrawals, and any serious events) will be communicated to the funder and oversight bodies per institutional requirements.

### Dissemination of trial findings

Results will be published in peer-reviewed journals and presented at scientific meetings. The full protocol and statistical analysis plan will be made publicly available (e.g., ClinicalTrials.gov or institutional repository).

### Trial oversight and monitoring

The trial steering committee comprises the trial principal investigators (NMA and MIAJ). It is responsible for the design, execution, and overall monitoring of the progress of the trial. Besides, the trial steering committee is also responsible for executing any trial modifications, including the ones proposed by the Data Monitoring Committee (DMC).

On the other hand, the TMG is responsible for ensuring a smooth conduct of the day-to-day trial operations. This comprises NMA (principal investigator) and other co-investigators (HA, IN, YJ, NG, JA and YSQ).

The DMC is appointed to monitor the trial safety data and comprises one independent statistician and a colorectal surgeon. All DMC members declared no conflict of interest.

### Trial status

At the time of submission, this clinical trial is still ongoing. Participant recruitment began on 01/04/2025 and was originally expected to be completed by 31/07/2025. Data collection was originally anticipated to be finalized by 15/08/2025 after the last check has been carried out to ensure data integrity. The study results were originally expected to be available by 30/09/2025. No interim analyses have been conducted so far, and no preliminary results have been generated or disseminated through any means of scientific publications or any form of public media. At present, the trial has just finished participant recruitment on 02/09/2025 and data collection is now projected to be finalized by 17/10/2025. As a result, the study results are now expected to available in full by 01/12/2025 (a delay of 2 months from the original timeline).

### Discussion

Effective bowel preparation is key to a high-quality colonoscopy. By leveraging AI, we aim to enhance patient education in a scalable way. The primary outcome, the BBPS, is a validated measure of bowel cleanliness [39,40], with higher scores reflecting clearer mucosa. Even modest improvements in BBPS can increase adenoma detection and reduce repeat procedures. Secondary outcomes (anxiety and satisfaction) address patient-centered effects: reduced anxiety and higher satisfaction may improve overall preparation adherence and experience. To avoid alpha-inflation, the protocol designates BBPS as the sole primary endpoint for confirmatory non-inferiority testing, whereas anxiety and satisfaction are secondary and interpreted descriptively [52,53,62]. Our findings will thus complement a recent AI-to-image concordance study in which ChatGPT-4 graded colonoscopy frames for BBPS and showed lower adequacy calls with slight agreement relative to expert raters [18]. In contrast, our trial evaluates patient-facing counseling as the intervention where bowel cleanliness is adjudicated by masked endoscopists rather than by the chatbot.

This trial incorporates rigorous methods to ensure reliability and safety. Masking of endoscopists and validated scales (BBPS, DASS-21, PSQ-18) promote unbiased outcome assessment. The sample size is powered for the expected effect on BBPS, and analyses will adjust for potential confounders.

Besides, the trial design and reporting align with internationally recognized clinical trial standards, thereby supporting methodological rigor and reproducibility for AI-based interventions. We have explicitly described the AI model version, input requirements, handling of bad input, human-AI interaction, and performance monitoring. We will analyze any AI performance errors and implement an escalation pathway if needed. Strategies to prevent hallucinations and bias include restricting content to evidence-based guidelines and auditing outputs. Privacy protections (no PII, encryption, access controls) safeguard patient data.

## Study limitations

The trial methodology has several limitations. First, apart from the endoscopists who are masked, both participants and the staff facilitating counseling or questionnaire administration are necessarily unmasked. This design constraint arises because the nature of the intervention (AI chatbot vs. clinician counseling) is inherently perceptible to patients, and staff members must directly support the delivery of each counseling modality. However, the risk of bias from unblinded staff is minimized by strict adherence to the trial protocol: counseling in the control arm follows standard hospital protocols, and questionnaire administration in both arms is performed by trained research assistants using verbatim scripted instructions, with responses entered electronically by patients themselves. Most importantly, the primary outcome (BBPS) is assessed objectively by masked endoscopists, ensuring that the central efficacy measure remains unbiased.

Second, requiring basic digital literacy and fluency in English or Malay may limit generalizability of our study findings. This first-phase trial intentionally restricts inclusion to participants able to engage with the intervention so that safety and fidelity of digital health-based intervention administration can be established without technical confounders [66–68]. Future work will adapt the system for broader populations (multilingual versions, voice interfaces, assisted navigation) and test effectiveness in individuals with limited digital or health literacy.

Third, we excluded participants with diagnosed clinical anxiety disorders to minimize confounding from psychiatric comorbidities and concurrent pharmacotherapy since both of which could mask or exaggerate the intervention's effect on situational procedural anxiety, thus limiting the generalizability of our trial findings. Given the modest sample size, inclusion of such patients also risked an imbalance between study arms that could distort outcomes. This approach is consistent with methodological practice in peri-procedural anxiety research, including colonoscopy studies, but it limits applicability to patients with baseline psychiatric diagnoses. Future studies should address this subgroup specifically, as they may derive distinct benefits from tailored interventions.

Fourthly, we employed ChatGPT 4.0 as the chatbot for delivering bowel preparation instructions for colonoscopy. With the release of ChatGPT 5.0 in mid-August 2025, our trial findings may appear outdated, as they do not reflect the performance of the most recent version. However, migrating to ChatGPT 5.0 during the trial is not feasible, since it would require major protocol amendments. Patients who have already received instructions using version 4.0 would need to be excluded, and a completely new cohort of participants would have to be recruited to ensure consistency with the updated version. We recognize this limitation, and recommend that future studies use ChatGPT 5.0 to evaluate and compare its effectiveness in delivering bowel preparation instructions against standard counseling provided by trained medical personnel.

Finally, we acknowledge that both first-time and repeat patients will be enrolled under simple randomization without a stratification scheme, hence introducing some degree of bias due to prior participants' prior colonoscopy experience. We will address this analytically by including it as a covariate and reporting subgroup analyses. This approach mitigates confounding risk, though a small degree of residual imbalance cannot be entirely excluded [63,64].

## Conclusion

If successful, this study will demonstrate that an AI chatbot can be a viable adjunct to traditional counseling, improving colonoscopy preparation quality. It will also provide a model for the transparent reporting of AI interventions in clinical trials, per SPIRIT-AI. Further research will be needed to generalize findings to other settings, but this trial aims to set a new standard for integrating AI safely into patient education.

## Supporting information

**S1 Table. The filled-in SPIRIT 2025 Checklist.**
(DOC)

**S1 File. Full trial protocol (AI Chatbot).**
(PDF)

## Acknowledgments

We thank the HCTM's staff and patients for their support and encouragement.

## Author contributions

**Conceptualization:** Nabil Mohammad Azmi, Muhammad Irfan Abdul Jalal.

**Funding acquisition:** Nabil Mohammad Azmi.

**Investigation:** Muhammad Irfan Abdul Jalal, Siti Hamizah Mohd Ashar, Muhammad Irfan Mohd Nazri, Young Jie, Nagulan Ganeson, Joane K Augustine.

**Methodology:** Nabil Mohammad Azmi, Muhammad Irfan Abdul Jalal, Siti Hamizah Mohd Ashar, Muhammad Irfan Mohd Nazri, Young Jie, Nagulan Ganeson, Joane K Augustine, Yew Sheng Qian.

**Project administration:** Nabil Mohammad Azmi, Muhammad Irfan Abdul Jalal, Siti Hamizah Mohd Ashar, Muhammad Irfan Mohd Nazri, Young Jie, Nagulan Ganeson, Joane K Augustine.

**Supervision:** Nabil Mohammad Azmi.

**Validation:** Muhammad Irfan Abdul Jalal.

**Writing – original draft:** Nabil Mohammad Azmi, Muhammad Irfan Abdul Jalal, Siti Hamizah Mohd Ashar, Muhammad Irfan Mohd Nazri, Young Jie, Nagulan Ganeson, Joane K Augustine, Yew Sheng Qian.

**Writing – review & editing:** Nabil Mohammad Azmi, Muhammad Irfan Abdul Jalal, Siti Hamizah Mohd Ashar, Muhammad Irfan Mohd Nazri, Young Jie, Nagulan Ganeson, Joane K Augustine, Yew Sheng Qian.

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
