## [Decision Letter · Decision Letter 0]

13 Aug 2025

Dear Dr. Abdul Jalal,

Thank you for submitting your manuscript to PLOS ONE. After careful consideration, we feel that it has merit but does not fully meet PLOS ONE’s publication criteria as it currently stands. Therefore, we invite you to submit a revised version of the manuscript that addresses the points raised during the review process.

We look forward to receiving your revised manuscript.

Kind regards,

Chih-Wei Tseng

Academic Editor

PLOS ONE

Journal Requirements: 

 [This study is financially sponsored by the Faculty of Medicine Fundamental Grant, Universiti Kebangsaan Malaysia (Project code: FF-2025-112).]. 

Reviewers' comments:

Reviewer's Responses to Questions

**Comments to the Author**

1. Does the manuscript provide a valid rationale for the proposed study, with clearly identified and justified research questions?

Reviewer #1: Yes

Reviewer #2: Partly

Reviewer #3: Yes

2. Is the protocol technically sound and planned in a manner that will lead to a meaningful outcome and allow testing the stated hypotheses?

Reviewer #1: Yes

Reviewer #2: Partly

Reviewer #3: Yes

3. Is the methodology feasible and described in sufficient detail to allow the work to be replicable?

Reviewer #1: Yes

Reviewer #2: No

Reviewer #3: Yes

4. Have the authors described where all data underlying the findings will be made available when the study is complete?

Reviewer #1: Yes

Reviewer #2: Yes

Reviewer #3: Yes

5. Is the manuscript presented in an intelligible fashion and written in standard English?

Reviewer #1: Yes

Reviewer #2: Yes

Reviewer #3: Yes

You may also provide optional suggestions and comments to authors that they might find helpful in planning their study.

Reviewer #1: This is a well-designed study protocol describing a single-masked, randomized controlled non-inferiority trial investigating the use of a ChatGPT-based AI chatbot to improve bowel preparation quality (BBPS) before colonoscopy. The topic is relevant and of growing interest, especially with the increasing integration of LLMs in patient education. The authors provide a detailed description of the study design, intervention, and statistical methods, following SPIRIT-AI guidelines.

However, while the protocol is methodologically robust, several clarifications and improvements are needed to strengthen the protocol prior to trial implementation.

Major Comments

1. While the manuscript clearly defines the statistical rule for non-inferiority (i.e., the lower bound of the 95% CI for BBPS difference must be greater than −1.1), the choice of the −1.1 BBPS point margin requires more clinical justification. The authors should elaborate why this value is considered clinically acceptable—e.g., by referencing the minimal clinically important difference (MCID), BBPS system validation, or its expected impact on detecting adenomas or determining adequacy (BBPS ≥6).

2. The study is single-masked (endoscopist only), but the participants and staff interacting with them are unblinded. This introduces a risk of performance and detection bias, particularly in patient-reported outcomes (anxiety, satisfaction). Please discuss how this potential bias will be mitigated (e.g., standardizing questionnaire delivery).

3. The inclusion of only digitally literate patients introduces a selection bias and limits generalizability, especially since poor bowel preparation is common among older adults, who may lack such literacy. This limitation should be more clearly acknowledged and discussed in the protocol.

4. While the manuscript outlines general plans for error logging and clinician review of 10% of sessions, it would benefit from specifying a threshold for unacceptable error rates that would trigger protocol revision or safety alerts.

5. It is unclear how adherence to bowel preparation instructions will be objectively assessed and compared between groups, especially since the chatbot allows for personalized interaction. Please define operational criteria for adherence in each arm (e.g., completion of laxative, compliance with timing).

Minor Comments

1. The manuscript inconsistently refers to BBPS as a non-inferiority outcome and sometimes as a superiority outcome. Please clarify this in both abstract and methods.

2. The manuscript refers to SPIRIT-AI in multiple sections in nearly identical wording. Consider reducing repetition for conciseness.

Reviewer #2: The authors of this manuscript—a trial protocol—present an interesting study, but I have several comments to offer.

1. To assert in the Introduction that no data exist on the use of ChatGPT for bowel preparation in colonoscopy is unsustainable. A study has already been published that yielded preliminary results through a concordance analysis (https://doi.org/10.3390/diagnostics14222537). The authors have neither cited nor discussed this work, which is presently the only one indexed in MEDLINE. Although that study employed an LLM to evaluate the BBPS rather than to provide information, it should nevertheless be cited to demonstrate how ChatGPT recognises the importance of bowel preparation for colonoscopy.

2. The study design and statistical plan describe a non-inferiority framework, yet the study is introduced as a “superiority randomised controlled trial”. This inconsistency must be corrected, and the primary hypothesis to be tested should be clearly defined.

3. The protocol lists BBPS, DASS-21 and PSQ-18 all as primary endpoints—three in total. How will multiplicity be handled? The resulting inflation of the alpha error will be considerable and must be discussed and justified; the sample-size calculation will presumably be greatly affected.

4. A two-sided test on a mean difference of 1.1 BBPS points is specified, but in a non-inferiority setting a one-sided test with H₀ defined on the delta margin is required. The delta of −1.1 should be based on solid clinical references (guidelines or correlation studies) and these should be cited, so that a loss of 1.1 points can be shown not to worsen the ADR in a clinically meaningful way, as well as remaining statistically coherent.

5. Does the planned sample size retain adequate power for the secondary outcomes as well?

6. The primary analysis proposes a t-test followed by a Mann–Whitney test, yet ANCOVA would be more appropriate, providing a 95 % confidence interval to be compared with delta in a non-inferiority context. Moreover, with only 96 participants you intend to adjust for seven covariates, entailing a marked risk of over-fitting; likewise, imputing 20 datasets in so small a sample carries a high likelihood of spurious variance.

7. The DASS-21 measures “sub-acute” anxiety and thus does not capture the immediate anxiety on the day of the examination, as it refers to the preceding week; similarly, the PSQ-18 is administered post-procedure, so post-sedative symptoms may influence it. These limitations need to be discussed.

8. Why exclude patients with clinical anxiety? Anxiety is an endpoint, and such patients are precisely those who might benefit from the intervention you propose. What will the findings mean in a cohort already calm at baseline, and to whom will they be applicable?

9. ChatGPT is not open source, contrary to the statement made in the manuscript.

10. Is it feasible to recruit patients meeting all these criteria within a few months without adopting a multicentre design?

11. Why employ an oncological CTCAE for diagnostic colonoscopies?

Reviewer #3: This study details the protocol for a prospective, randomized controlled trial employing a non-inferiority framework. The study is designed to compare the efficacy of a novel ChatGPT-based counselling intervention with standard-of-care counseling for patients undergoing bowel preparation for colonoscopy. The trial will enroll 96 participants at a single center, with allocation to the intervention or control arm in a 1:1 ratio. The primary endpoint is the Boston Bowel Preparation Score (BBPS). Secondary endpoints will assess patient-reported outcomes, including anxiety via the DASS-21 scale and satisfaction using the PSQ-18. The protocol has secured ethics approval and adheres to both SPIRIT-2025 and the specialized SPIRIT-AI reporting standards.

Major concerns:

1. Line 137-138: One aspect of your eligibility criteria allows both first‑time and repeat elective colonoscopy patients to enroll. Patients who have previously undergone colonoscopy may already be familiar with dietary restrictions and bowel‑preparation instructions, potentially affecting their baseline knowledge, anxiety and adherence compared with first‑time patients. How do you plan to account for this difference in your study design or analysis? Have you considered stratifying randomization by prior colonoscopy experience or adjusting for it in your statistical models to ensure that the intervention’s effect is not confounded by participants’ prior familiarity with the procedure?

2. Line 162-163: The protocol indicates that outcome assessments will be performed by experienced endoscopists who are blinded to the intervention. However, the manuscript does not specify how many endoscopists will participate in the trial or how inter‑observer variability will be managed. Could you clarify the number of endoscopists involved and describe any training or calibration procedures to ensure consistent scoring of the BBPS?

3. Line 180-189: In describing the intervention, the authors state that the chatbot’s content is based on guidelines from the ESGE and the MOH. However, the laxative timing and dosage in your protocol are slightly different from the 2019 ESGE recommendations. Could you specify which year/version of the ESGE guidelines you followed and explain any deviations in timing or dosage? How did you ensure that your regimen remains evidence‑based given these differences?

4. Line 244-250: Given that your intervention relies on a proprietary OpenAI model that is periodically updated, how will you ensure that participants receive consistent guidance throughout the trial? Your protocol mentions documenting model updates and assessing consistency but does not specify whether the ChatGPT version will be frozen for the duration of the study. Will you lock in a specific model version and settings, or implement procedures to detect and mitigate any behavioral changes if OpenAI releases an update mid‑trial?

5. Line 251-256: Your protocol states that patients will interact directly with the ChatGPT‑based chatbot without real‑time expert oversight and that the model will operate with a temperature parameter of 1.0 to encourage “creative and empathetic responses”. This raises the potential concern of high‑temperature settings can increase the likelihood of ambiguous or inaccurate advice. Could you justify the choice of a temperature setting of 1.0 and explain whether lowering it to reduce variability was considered?

6. Line 257-274: While the protocol details several mechanisms for classifying and logging AI‑generated errors and for escalating clinically inappropriate responses to human counselling, it does not define what level of error would be considered unacceptable or what corrective actions would follow. Could you clarify whether you have set quantitative thresholds for error rates or types that would trigger modifications to the chatbot prompts or suspension of the intervention? Additionally, how will decisions be made about implementing changes to the chatbot’s content if persistent errors are identified during monitoring?

7. Your eligibility criteria favor participants with “basic digital literacy” and fluency in English or Malay, and exclude those with cognitive impairment or complex comorbidities. This means the trial will largely involve individuals who are already comfortable with technology and able to understand health instructions, whereas those with low health literacy, limited language skills or complex medical needs—often the patients who have the most difficulty preparing for colonoscopy—will not be included. Given these inclusion criteria, could the authors comment on the selection bias and whether the study population truly reflects the broader patient groups most in need of improved bowel‑preparation support?

Minor concerns:

1. "Dr(Surgery)" for Nabil Mohammad Azmi: This is an uncommon degree abbreviation. Please check.

2. Line 203: “No PII…..” is at first use. Line 244 “PII” has full term. The abbreviation should be defined upon its first use. Please write out the full term followed by the acronym in parentheses.

3. Line 220: “will be assessed using the “Flesh Kincaid” Grade Level score” should be “Flesch Kincaid”.

4. Line 395: should be “TMG”.

5. The manuscript uses both “counselling” and “counseling.” Please review the text for consistency and adopt a single preferred form throughout.

6. Line 535-536: "The primary outcome, the BBPS, is a validated measure of bowel cleanliness [18]": Reference [18] is for a psychiatric study using Woebot (JMIR Ment Health. 2017). The correct reference for BBPS validation would be [25] Lai EJ et al. (Gastrointest Endosc. 2009).

7. Reference 7 and 10 appear to be duplicates.

**Do you want your identity to be public for this peer review?** For information about this choice, including consent withdrawal, please see our Privacy Policy

Reviewer #1: No

Reviewer #2: No

Reviewer #3: No

---

## [Author Response · Author response to Decision Letter 1]

10 Sep 2025

A) Reviewer 1: Detailed Point-by-Point Responses

1) Reviewer’s Major Comment 1: Clinical justification for -1.1 BBPS non-inferiority margin

While the manuscript clearly defines the statistical rule for non-inferiority (i.e., the lower bound of the 95% CI for the BBPS difference must be greater than -1.1), the choice of the -1.1 BBPS point margin requires more clinical justification. The authors should elaborate why this value is considered clinically acceptable, e.g., by referencing the minimal clinically important difference (MCID), BBPS system validation, or its expected impact on detecting adenomas or determining adequacy (BBPS ≥6).

i) Author’s Response:

We appreciate the reviewer’s observation and agree that the justification for the non-inferiority margin should be grounded in robust clinical evidence. The -1.1-point non-inferiority margin for the Boston Bowel Preparation Score (BBPS) was selected based on published literature assessing the minimal clinically important difference (MCID) for the Boston Bowel Preparation Scale (BBPS) and its correlation with adenoma detection rate (ADR). Lai et al. validated the BBPS and found that adequate preparation, defined as a total BBPS score ≥6 with no segment <2, is associated with acceptable ADRs and caecal intubation rates [39]. Calderwood et al. demonstrated that differences of ≤1 point across BBPS total scores are not associated with clinically meaningful differences in ADR or polyp detection [40].

Furthermore, a non-inferiority margin of -1.1 BBPS points preserves the probability of maintaining an adequate bowel preparation rate above the 85% benchmark recommended by the European Society of Gastrointestinal Endoscopy (ESGE), since even after allowing for this margin the expected mean BBPS remains above the adequacy threshold (≥6) [25]. For example, if the control group mean is 7.5, applying the -1.1 BBPS margin still yields 6.4, which corresponds to at least 85% of patients achieving adequate bowel preparation. Setting a stricter margin would risk underpowering the study unnecessarily, whereas a wider margin could compromise clinical relevance. Our choice is therefore both statistically coherent and clinically conservative and meaningful.

ii) Exact manuscript change: In Methods (Subsection Statistical Analysis)

“The non-inferiority margin of -1.1 BBPS points was selected based on validation studies showing that differences of this magnitude do not compromise adenoma detection rate or adequacy classification (BBPS ≥6 with all segments ≥2) [39, 40]. This threshold is also consistent with ESGE guideline quality benchmarks that require at least 85% adequacy in bowel preparation [23].” (pages 31-32; lines 595 - 600)

iii) References (citation order and number are as in the main manuscript):

23. Hassan C, East J, Radaelli F, Spada C, Benamouzig R, Bisschops R, et al. Bowel preparation for colonoscopy: ESGE Guideline - Update 2019. Endoscopy. 2019;51(8):775–794. doi:10.1055/a-0959-0505.

39. Lai EJ, Calderwood AH, Doros G, Fix OK, Jacobson BC. The Boston bowel preparation scale: a valid and reliable instrument for colonoscopy-oriented research. Gastrointest Endosc. 2009;69(3 Pt 2):620-5. doi: 10.1016/j.gie.2008.05.057.

40. Calderwood AH, Schroy PC 3rd, Lieberman DA, Logan JR, Zurfluh M, Jacobson BC. Boston Bowel Preparation Scale scores provide a standardized definition of adequate for describing bowel cleanliness. Gastrointest Endosc. 2014;80(2):269–276. doi: 10.1016/j.gie.2014.01.031.

2) Reviewer’s major Comment 2: Risk of performance/detection bias with single masking

The study is single-masked (endoscopist only), but the participants and staff interacting with them are unblinded. This introduces a risk of performance and detection bias, particularly in patient-reported outcomes (anxiety, satisfaction). Please discuss how this potential bias will be mitigated (for instance, standardizing questionnaire delivery).

i) Author’s Response:

We thank the reviewer for raising this important point. We acknowledge that in a single-masked trial, both participants and staff interacting with them are unblinded, which may raise concerns about performance and detection bias in patient-reported outcomes (anxiety, satisfaction). To mitigate this, we have implemented multiple safeguards, consistent with SPIRIT, CONSORT, and Cochrane guidance:

1. Standardized administration – All patient-reported outcomes (DASS-21 and PSQ-18) will be delivered by research assistants using a verbatim scripted protocol, independent of allocation.

2. Electronic self-completion – Patients will enter responses directly into a secure electronic system (REDCap®), eliminating interviewer influence and transcription errors.

3. Training and monitoring – Research assistants will undergo structured training, competency assessment, and ongoing monitoring. The trial coordinator will conduct random audits to ensure fidelity.

4. Temporal and environmental standardization – Questionnaires will be administered at fixed, prespecified time points (immediately pre- and post-colonoscopy) in a standardized quiet environment.

5. Validated tools – Both DASS-21 and PSQ-18 are validated in Malay and English, with high reliability, reducing susceptibility to contextual variation.

Regarding patient blinding, this is not feasible in our setting. The intervention being compared is AI chatbot counselling versus standard human counselling, and participants are necessarily aware of which modality they are receiving. Concealing this would be impractical, as patients can readily recognize whether they are interacting with an AI chatbot or a clinician. Thus, any attempt to conceal this would be artificial and unworkable.

Regarding staff blinding, only outcome assessors (endoscopists) are blinded, since they evaluate the primary endpoint (BBPS) during colonoscopy. Other staff members (e.g., study recruiters, research assistants administering questionnaires) cannot be blinded due to their direct role in facilitating either the chatbot session or the human counselling. However, their influence on outcomes is minimized by rigorous adherence to the trial protocol:

• Staff delivering counselling in the control arm follow a standardized hospital protocol.

• Research assistants administering questionnaires are not involved in providing counselling, and their interaction is limited to reading scripted instructions verbatim.

• Staff are trained to avoid non-verbal cues or discussion that could bias patient perceptions.

Therefore, while participants and some staff are necessarily unblinded, the measures above minimize the risk of systematic differences between trial arms. More importantly, the primary endpoint (BBPS) is objectively assessed by endoscopists who remain masked, ensuring the principal outcome of the trial is unbiased.

ii) Exact manuscript changes:

a) In Methods (Subsections Trial Endpoint Assessments & Randomization and Masking)

- “To minimize performance and detection bias, patient-reported outcomes (DASS-21, PSQ-18) will be administered by trained research assistants using a verbatim scripted protocol, without disclosure of allocation. Patients will complete questionnaires electronically, ensuring self-entry and eliminating interviewer influence. Research assistants will receive structured training and ongoing monitoring to ensure protocol fidelity. All questionnaires will be administered at prespecified time points in a quiet, standardized environment.” (Page 22-23; Lines 422-428)

- “Complete masking of participants and staff is not feasible since participants are inherently aware of whether they interact with an AI chatbot or a clinician, and staff facilitating the counseling or questionnaires necessarily know the intervention delivered. However, staff influence will be minimized through strict adherence to standardized hospital counseling protocols (control arm) and scripted questionnaire delivery (ChatGPT arms). Importantly, the BBPS is objectively evaluated by endoscopists who are masked to allocation, preserving the integrity of the primary endpoint evaluation. The endoscopists and participants will be instructed to maintain minimal conversation to prevent unmasking. No emergency unmasking is anticipated since we do not expect any significant harm to be associated with the interventions received by the participants.” (Pages 14-15; Lines 179-189)

b) Discussion (Subsection Statistical Analysis)

- “First, apart from the endoscopists who are masked, both participants and the staff facilitating counseling or questionnaire administration are necessarily unmasked. This design constraint arises because the nature of the intervention (AI chatbot vs. clinician counseling) is inherently perceptible to patients, and staff members must directly support the delivery of each counseling modality. However, the risk of bias from unblinded staff is minimized by strict adherence to the trial protocol: counseling in the control arm follows standard hospital protocols, and questionnaire administration in both arms is performed by trained research assistants using verbatim scripted instructions, with responses entered electronically by patients themselves. Most importantly, the primary outcome (BBPS) is assessed objectively by masked endoscopists, ensuring that the central efficacy measure remains unbiased.” (Page 36; Lines 705-716)

3) Reviewer’s Major Comment 3: The inclusion of only digitally literate patients and effects on selection bias and generalizability issues

The inclusion of only digitally literate patients introduces a selection bias and limits generalisability, especially since poor bowel preparation is common among older adults, who may lack such literacy. This limitation should be more clearly acknowledged and discussed in the protocol.

i) Author’s Response:

We thank the reviewer for this important point. We acknowledge that restricting eligibility to participants with basic digital literacy and fluency in English or Malay may limit generalizability, particularly for older adults and those with reduced technology access, who are also populations at risk of inadequate bowel preparation.

This restriction was a deliberate and pragmatic choice for a first-in-context safety and feasibility trial. Early-phase evaluations of digital health and AI interventions commonly begin with populations capable of reliably engaging with the technology, in order to establish internal validity and safeguard content fidelity and safety before extending to broader populations [68-70].

To mitigate exclusion, participants without personal devices but otherwise eligible were provided with loaner tablets and a short onboarding session (≤10 minutes). Baseline digital literacy and language proficiency are recorded in the case report form, and these distributions will be reported descriptively in the results. We do not plan inferential subgroup analyses of digital literacy strata given the modest sample size, but transparent reporting will allow readers to judge representativeness. Future work will adapt the chatbot for low-literacy and non-English/Malay speakers through simplified interfaces, voice-based versions, and multilingual expansion.

ii) Exact manuscript changes:

a) In Methods (Subsection Eligibility Criteria)

“We require participants to have basic digital literacy in order to interact with the chatbot interface. For participants who are otherwise eligible but lack a personal device, the trial will provide a loaner tablet and a short onboarding session (10 minutes or less) to enable participation. The provision of loaner devices aims to mitigate but not eliminate selection bias; the trial remains a pragmatic first-phase evaluation among individuals who can use a mobile/tablet interface.” (Page 13; Lines 149-155)

b) In Discussion (Subsection Study Limitations)

Second, requiring basic digital literacy and fluency in English or Malay may limit generalizability of our study findings. This first-phase trial intentionally restricts inclusion to participants able to engage with the intervention so that safety and fidelity of digital health-based intervention administration can be established without technical confounders [68-70]. Future work will adapt the system for broader populations (multilingual versions, voice interfaces, assisted navigation) and test effectiveness in individuals with limited digital or health literacy.” (Pages 36 -37; Lines 717 – 723)

iii) References (Citation order is as in the main manuscript)

68. Marcolino MS, Oliveira JAQ, D’Agostino M, et al. The impact of mHealth interventions: systematic review of systematic reviews. JMIR Mhealth Uhealth. 2018;6(1):e23.

69. Mohr DC, Weingardt KR, Reddy M, Schueller SM. Three problems with current digital mental health research… and three things we can do about them. Psychiatr Serv. 2017;68(5):427–429.

70. Whittaker R, Merry S. Technology-dissemination and digital health intervention frameworks. Lancet Digital Health. 2021;3:e2–e3.

4) Reviewer’s Major Comment 4: Threshold for Unacceptable error rates

While the manuscript outlines general plans for error logging and clinician review of 10% of sessions, it would benefit from specifying a threshold for unacceptable error rates that would trigger protocol revision or safety alerts.

i) Author’s response:

We thank the reviewer for this important suggestion. We agree that pre-defined thresholds are essential for transparent AI safety governance. In response, we have expanded the protocol to include a quantitative, risk-based error taxonomy and escalation plan with explicit safety triggers:

• Error taxonomy (pre-specified):

Critical error: AI output that could directly cause patient harm (e.g., incorrect dosing interval that risks electrolyte imbalance, or wrong instructions about fasting before sedation). Any critical error triggers immediate suspension of the chatbot for that pathway and urgent review by the trial Safety Officer and Data Monitoring Committee (DMC).

Major error: Incorrect advice that could reduce bowel preparation adequacy (e.g., instructing a patient to take the final laxative >12 hours before procedure). These require prompt corrective action and expanded audit.

Minor error: Non-clinical issues (typos, ambiguous phrasing) that do not affect safety or efficacy, but still require content refinement if recurrent.

• Quantitative thresholds and actions (pre-specified):

Immediate suspension: Any single critical error in an audited transcript will result in immediate suspension of the chatbot pathway and urgent DMC review.

Prompt corrective action: Major errors ≥1% of audited sessions in a rolling 2-week window for root-cause analysis, prompt correction, and expanded audit (25% sample) until rates fall below threshold.

Content refinement: Minor errors ≥5% across two consecutive audits will lead to content editing and template re-training, followed by monitoring.

• Audit and monitoring schedule:

Weekly automated error logging; clinician review of a random 10% of transcripts (escalated to 25% if thresholds are breached); and monthly safety summaries to the DMC. All transcripts contain version metadata, ensuring that any drift after model updates can be traced.

This approach is consistent with best practices from WHO [1], SPIRIT-AI/CONSORT-AI [2], BMJ safety frameworks [3], the Topol Review [4], NHSX policy [5], the NIST AI Risk Management Framework [6], and explainability scholarship in healthcare AI [7].

ii) Exact Manuscript Changes:

a) “AI error taxonomy, thresholds and escalation

All chatbot interactions will be logged and reviewed according to a pre-specified error taxonomy. Errors are classified as Critical, Major or Minor (definitions below). A weekly random sample of 10% of sessions will be reviewed by trained clinicians and review sampling will expand to 25% if thresholds are exceeded. Escalation rules are:

1. Critical error: any instance of advice that could cause direct physical harm (e.g., wrong dosing interval for th

---

## [Decision Letter · Decision Letter 1]

26 Sep 2025

A Prospective Single-Masked, Non-Inferiority, Parallel-Group Randomized Controlled Trial of the Efficacy of a ChatGPT-Based AI Chatbot to Improve Boston Bowel Preparation Scores for Colonoscopy Preparation: A Trial Protocol

PONE-D-25-33008R1

Dear Dr. Abdul Jalal,

We’re pleased to inform you that your manuscript has been judged scientifically suitable for publication and will be formally accepted for publication once it meets all outstanding technical requirements.

Kind regards,

Chih-Wei Tseng

Academic Editor

PLOS ONE

Additional Editor Comments (optional):

Reviewers' comments:

Reviewer's Responses to Questions

**Comments to the Author**

1. Does the manuscript provide a valid rationale for the proposed study, with clearly identified and justified research questions?

Reviewer #1: Yes

Reviewer #2: Yes

Reviewer #3: Yes

2. Is the protocol technically sound and planned in a manner that will lead to a meaningful outcome and allow testing the stated hypotheses?

Reviewer #1: Yes

Reviewer #2: Yes

Reviewer #3: Yes

3. Is the methodology feasible and described in sufficient detail to allow the work to be replicable?

Reviewer #1: Yes

Reviewer #2: Yes

Reviewer #3: Yes

4. Have the authors described where all data underlying the findings will be made available when the study is complete?

Reviewer #1: Yes

Reviewer #2: Yes

Reviewer #3: Yes

5. Is the manuscript presented in an intelligible fashion and written in standard English?

Reviewer #1: Yes

Reviewer #2: Yes

Reviewer #3: Yes

You may also provide optional suggestions and comments to authors that they might find helpful in planning their study.

Reviewer #1: The authors have fully addressed previous concerns. The manuscript is clear, consistent, and suitable for acceptance.

Reviewer #2: The authors revised the manuscript concerning the remaining issues. The new version is now improved. Congratulations.

Reviewer #3: The authors have been thorough and responsive, satisfactorily addressing all the major and minor concerns raised in my initial review. The resulting changes have substantially strengthened the manuscript.

**Do you want your identity to be public for this peer review?** For information about this choice, including consent withdrawal, please see our Privacy Policy

Reviewer #1: No

Reviewer #2: No

Reviewer #3: No

---

## [Editor Report · Acceptance letter]

PONE-D-25-33008R1

PLOS ONE

Dear Dr. Abdul Jalal,

I'm pleased to inform you that your manuscript has been deemed suitable for publication in PLOS ONE. Congratulations! Your manuscript is now being handed over to our production team.

Kind regards,

on behalf of

Dr. Chih-Wei Tseng

Academic Editor

PLOS ONE